# Occupational Exposure to Inhalational Anesthetics and Teratogenic Effects: A Systematic Review

**DOI:** 10.3390/healthcare11060883

**Published:** 2023-03-17

**Authors:** José Manuel García-Álvarez, Guillermo Escribano-Sánchez, Eduardo Osuna, Alonso Molina-Rodríguez, José Luis Díaz-Agea, Alfonso García-Sánchez

**Affiliations:** 1Health Sciences PhD Program, Campus de los Jerónimos n°135, Universidad Católica de Murcia UCAM, Guadalupe, 30107 Murcia, Spain; 2Faculty of Nursing, Catholic University of Murcia, Guadalupe, 30107 Murcia, Spain; 3Departamento de Ciencias Sociosanitarias, Área de Medicina Legal y Forense, Universidad de Murcia, 30107 Murcia, Spain; 4Faculty of Nursing, Campus de Ciencias de la Salud, Edificio LAIB/DEPARTAMENTAL, University of Murcia, El Palmar-Murcia, 30120 Murcia, Spain

**Keywords:** anesthetic gases, inhalational anesthetics, abortion spontaneous, congenital abnormalities, teratogenesis

## Abstract

(1) Background: In the current healthcare environment, there is a large proportion of female staff of childbearing age, so, according to existing conflicting studies, the teratogenic effects that inhalational anesthetics may have on exposed pregnant workers should be assessed. This investigation aims to analyze the teratogenic effects of inhalational anesthetics in conditions of actual use, determining any association with spontaneous abortion or congenital malformations. (2) Methods: A systematic review was carried out according to the PRISMA statement based on PICO (problem of interest–intervention to be considered–intervention compared–outcome) (Do inhalational anesthetics have teratogenic effects in current clinical practice?). The level of evidence of the selected articles was evaluated using the SIGN scale. The databases used were PubMed, Embase, Scopus, Web of Science, Google academic and Opengrey. Primary studies conducted in professionals exposed to inhalational anesthetics that evaluate spontaneous abortions or congenital malformations, conducted in any country and language and published within the last ten years were selected. (3) Results: Of the 541 studies identified, 6 met all inclusion criteria in answering the research question. Since many methodological differences were found in estimating exposure to inhalational anesthetics, a qualitative systematic review was performed. The selected studies have a retrospective cohort design and mostly present a low level of evidence and a low grade of recommendation. Studies with the highest level of evidence do not find an association between the use of inhalational anesthetics and the occurrence of miscarriage or congenital malformations. (4) Conclusions: The administration of inhalational anesthetics, especially with gas extraction systems (scavenging systems) and the adequate ventilation of operating rooms, is not associated with the occurrence of spontaneous abortions or congenital malformations.

## 1. Introduction

Inhalational anesthetics are drugs widely used in patients undergoing surgery under general anesthesia. The relationship between their use and the development of teratogenic effects in exposed professionals is still under debate [1,2]. The first study on the adverse effects of inhalational anesthetics was that of Vaisman [3] in 1967, who analyzed the working conditions of anesthesiologists exposed to these substances and reported a direct association between exposure and spontaneous abortions in pregnant anesthesiologists. Subsequent studies also established a possible relationship between inhalational anesthetics and adverse effects during pregnancy capable of causing spontaneous abortions or congenital malformations in operating room nurses [4] or healthcare workers [5].

Some studies, conducted with laboratory animals, pointed to adverse effects during pregnancy that were associated with long periods of exposure to inhalational anesthetics at high concentrations, as well as coinciding with the period of organogenesis [6,7,8,9], conditions that are difficult to reproduce in humans. Research on their use in humans has detected some genetic alterations that may be potentially teratogenic [10,11,12,13,14], although a subsequent review failed to establish a relationship between inhalational anesthetics and genotoxic effects [15]. Some retrospective studies performed on health professionals of reproductive age exposed to inhalational anesthetics found an association between exposure and the occurrence of spontaneous abortions or congenital malformations, while other investigations failed to establish any relationship [1,16,17,18,19,20].

Several reviews admit the possibility of an association between occupational exposure to inhalational anesthetics and adverse pregnancy outcomes that could be manifested in spontaneous abortions or congenital malformations [21,22,23,24]. However, the validity of the findings has been questioned due to the methodological weaknesses of the studies analyzed in which problems in sample selection, low response rates, inadequate information on exposure levels or the existence of different confounding factors (antineoplastic drugs, sterilizing substances, radiation, tobacco or alcohol consumption, stressful situations, overwork, unregulated schedules, inadequate temperature and humidity conditions, etc.), which were not taken into account and could have interfered with the results obtained by being the possible cause of the problems detected, have been observed. Therefore, there is no high-quality evidence that demonstrates beyond doubt an association between adverse effects occurring during pregnancy and exposure to low levels of inhalational anesthetics [25,26,27].

One aspect to take into account is that most of the studies analyzing this problem were performed before the widespread application of gas extraction systems (scavenging systems). These systems would have considerably reduced the risk of overexposure to inhalational anesthetics in operating rooms and, therefore, their possible harmful effects on the health of pregnant healthcare professionals [28,29].

This investigation aims to analyze the possible relationship between occupational exposure to inhalational anesthetics and adverse teratogenic effects among female health professionals in current working conditions. Any association with the development of spontaneous abortions or congenital malformations is analyzed to establish the overall strengths and consistency of such associations.

## 2. Materials and Methods

In order to achieve this objective, a systematic review was conducted. This review was conducted in accordance with the Preferred Reporting Items for Systematic Reviews and Meta-Analyses (PRISMA) standards [30].

### 2.1. Criteria for Considering Studies

As the first step, the following research question was formulated to guide this systematic review using the PICO strategy (problem of interest–intervention to be considered–intervention with which to compare–result) [31]: Do inhalational anesthetics present teratogenic effects in current clinical practice?

For this review, we selected primary studies that answered the research question, that were conducted in female professionals exposed to inhalational anesthetics, that assessed the presence of spontaneous abortions or congenital malformations and that were published in any country or language during the last 10 years.

### 2.2. Search Strategies and Data Resources

A literature search was performed in PubMed, Embase, Scopus, Web of Science, Google academic and Opengrey in November 2022. The following search strategy was used to search each database: (“anesthetic gases” OR “anaesthetic gases” OR “inhalational anesthetics” OR “inhalational anaesthetics” OR “nitrous oxide” OR “halothane” OR “isoflurane” OR “sevoflurane” OR “desflurane” OR “enflurane”) AND (“abortion spontaneous” OR “congenital abnormalities” OR “teratogenesis” OR “teratology”).

### 2.3. Data Extraction

Data extraction was carried out using a method akin to screening. The studies obtained after the search in the different databases used were initially selected according to the title and abstract. Subsequently, these articles were then read in their entirety to select those that fully met the inclusion criteria. The following information was collected: title, author/s, journal and year of publication, country and language, objective, type of study, study population, method of data collection, type of inhalational anesthetic and results.

### 2.4. Quality Assessment

The Scottish Intercollegiate Guidelines Network (SIGN) scale was used to assess the quality of the included studies. The SIGN scale classifies studies according to the level of scientific evidence and the degree of recommendation. This scale was used because it gives more weight to observational studies, which are the only ones that, for ethical reasons, can be performed to investigate the teratogenic effects of inhalational anesthetics in humans [32].

### 2.5. Collecting, Summarizing and Reporting Results

The researchers used dichotomous qualitative outcome variables (yes/no exposure, yes/no spontaneous abortion, yes/no congenital malformations), so the summary measures are the number and percentage of patients exposed to the event of interest or the association measures, using the likelihood ratio (odds ratio).

The selected studies present numerous methodological differences, which explains why only a qualitative systematic review is carried out, where the results and characteristics of the individual studies are presented to facilitate the observation of similarities and differences between them, thus allowing for comparisons.

## 3. Results

The literature search retrieved 541 records. After removing 19 duplicate studies, we screened 522 abstracts for eligibility, eliminating 501 studies because reading the title and summary showed that they did not meet all the inclusion criteria. After the complete reading of the remaining 21 studies, 15 were excluded for not fulfilling 1 or more of the inclusion criteria (Figure 1). This meant that six were left for a detailed analysis, which is summarized in Table 1.

Study 1 [16]. Title: “Occupational exposures among nurses and risk of spontaneous abortion”. Authors: Lawson CC, Rocheleau CM, Whelan EA, Lividoti-Hibert EN, Grajewski B, Spiegelman D and Rich-Edwards JW. Journal and year of publication: *American Journal of Obstetrics and Gynecology*, 2012. Country and language: United States, English. Objective: To assess occupational exposure and the risk of miscarriage in nurses. Type of study: Retrospective cohort design. Study population: 116,430 US nurses. Data collection method: Biannual self-completed survey from 1993 to 2001 on work and reproductive history. Those who responded that they had worked during their last pregnancy (11,177 nurses) were sent a specific complementary questionnaire that inquired about different aspects such as exposure to inhalational anesthetics, sterilizing agents, antineoplastic drugs, antiviral drugs or radiation; work schedule; lifting weights; hours of standing or walking at work; physical exercise; and the consumption of tobacco, coffee or alcohol. The exposed cohort consisted of nurses who reported having been in contact with inhalational anesthetics for more than 1 h per day. Type of inhalational anesthetic: Halothane, isoflurane, enflurane and nitrous oxide. Results: A total of 89% of the nurses responded, and 7842 were selected who had been in contact with inhalational anesthetics. A total of 775 spontaneous abortions (10.4% of pregnancies) were reported. The following were associated with an increased risk of miscarriage: exposure to antineoplastic drugs, radiation or sterilizing agents; the consumption of coffee, alcohol or tobacco; increased working hours; and age. No statistically significant association was found between miscarriage and the use of inhalational anesthetics in exposed nurses. Level of evidence SIGN: 2+ because it is a cohort study with a low risk of confounding, bias or chance and a moderate probability that the relationship is not causal due to the fact that it quantifies exposure in terms of the length of stay in the operating room and not in terms of the actual dose of exposure to anesthetic gases. Grade of recommendation SIGN: C because the studies are consistent and directly applicable to the target population.

Study 2 [33]. Title: “Inhaled anesthetics and the reproductive risk associated with occupational exposure among female veterinary anesthesia workers”. Authors: Allweiler SI and Kogan LR. Journal and year of publication: *Veterinary Anaesthesia and Analgesia*, 2013. Country and language: United States, English. Objective: To assess the reproductive alterations produced by inhalational anesthetics in veterinary clinics. Type of study: Retrospective cohort design. Study population: 295 US veterinarians. Data collection method: Self-completed survey on work history and the occurrence of spontaneous abortions or congenital malformations. This study assessed exposure to inhalational anesthetics, stress, heavy lifting, working time and variable schedules. The exposed cohort consisted of 209 veterinarians working in the operating room with inhalational anesthetics, and the unexposed cohort consisted of 86 veterinarians working in intensive care. Type of inhalational anesthetic: Not specified. Results: The percentage of congenital malformations in the offspring of exposed veterinarians was 1.5%, and in those not exposed, it was 1.2%. The percentage of spontaneous abortions in exposed veterinarians was 12.4%, and in non-exposed veterinarians, it was 7.1%. No statistically significant association was observed between inhalational anesthetics and congenital malformations or spontaneous abortions. Level of evidence SIGN: 2+ because it is a cohort study with a low risk of confounding, bias or chance and a moderate probability that the relationship is not causal due to the fact that it quantifies exposure according to the workplace and not according to the real dose of exposure of each worker to inhalational anesthetics. Grade of recommendation SIGN: C because the studies are consistent and directly applicable to the target population.

Study 3 [34]. Title: “Anaesthesia practice and reproductive outcomes: Facts unveiled”. Authors: Nagella AB, Ravishankar M and Hemanth-Kumar VR. Journal and year of publication: *Indian Journal of Anaesthesia*, 2015. Country and language: Indian, English. Objective: To assess anesthetic practices and their relationship with reproductive disorders in anesthesiologists. Type of study: Retrospective cohort design. Study population: 9974 Indian anesthesiologists. Method of data collection: Self-completed survey on anesthetic technique used and the occurrence of spontaneous abortions or congenital malformations. The exposed cohort consisted of 207 female anesthesiologists who had worked in the operating room during the first trimester of pregnancy, and the unexposed cohort consisted of 138 female anesthesiologists who had worked outside the operating room during the first trimester of pregnancy. Type of inhalational anesthetic: Isoflurane, halothane, desflurane, sevoflurane and nitrous oxide. Results: A total of 1563 anesthesiologists (15.7%) responded. Eighty-five percent of the operating rooms analyzed in this study did not have gas extraction systems (scavenging systems). A higher risk of spontaneous abortions was observed in exposed patients. A statistically significant association was found between time worked and congenital malformations, but not between the type of inhalational anesthetics used and reproductive alterations. Level of evidence SIGN: 2- because it is a cohort study with a high risk of confounding, bias or chance and a significant probability that the relationship is not causal because the response rate is very low and does not take into account other occupational or lifestyle-related risk factors capable of causing these reproductive alterations. Grade of recommendation SIGN: Studies with this level of evidence should not be used in the process of making recommendations because of the high possibility of bias.

Study 4 [35]. Title: “Health Effects Associated with Exposure to Anesthetic Gas Nitrous Oxide-N_2_O in Clinical Hospital-Shtip Personel”. Authors: Eftimova B, Sholjakova M, Mirakovski D and Hadzi-Nikolova M. Journal and year of publication: *Open Access Macedonian Journal of Medical Sciences*, 2017. Country and language: Macedonia, English. Objective: To assess the effects of occupational exposure to nitrous oxide. Type of study: Retrospective cohort design. Study population: 43 Macedonian hospital workers. Data collection method: Self-completed survey on work history and the possible effects produced by nitrous oxide, including the occurrence of spontaneous abortions. Exposure was calculated by measuring nitrous oxide levels in operating rooms during an eight-hour working day using a portable electrochemical instrument for each worker. The exposed cohort consisted of 23 operating room and intensive care unit workers (20 women), and the unexposed cohort consisted of 20 internal medicine workers (16 women). Type of inhalational anesthetic: Nitrous oxide. Results: No increased risk of spontaneous abortion was found in the exposed cohorts. No statistically significant association was observed between operating room nitrous oxide concentrations and miscarriages. Level of evidence SIGN: 2++ because it is a high-quality cohort study with a very low risk of bias and a high probability of establishing a causal relationship because it quantifies the exact exposure of each worker to nitrous oxide. Grade of recommendation SIGN: B because it is a very consistent study and directly applicable to the target population.

Study 5 [36]. Title: “Occupational genotoxic effects in a group of nurses exposed to anesthetic gases in operating rooms of zagazia university hospitals”. Authors: Borayek GE, El-Magd SA, El-Gohary SS, El-Naggar AM and Hammouda MA. Journal and year of publication: *Egyptian Journal of Occupational Medicine*, 2018. Country and language: Egypt, English. Objective: To identify the genotoxic effects of anesthetic gases in operating room nurses. Type of study: Retrospective cohort design. Study population: 62 Egyptian hospital nurses. Data collection method: Self-completed survey on work history and occurrence of spontaneous abortions or congenital malformations. Urine isoflurane concentration was determined, and chromosomal alterations were assessed using blood cell karyotyping. The exposed cohort consisted of 32 operating room nurses, and the unexposed cohort consisted of 32 hospital outpatient nurses. Type of inhalational anesthetic: Isoflurane. Results: Most of the operating rooms analyzed did not have a gas extraction system (a scavenging system). A higher risk of chromosomal alterations, spontaneous abortions and congenital malformations was observed in the exposed nurses. No statistically significant association was found between urine isoflurane levels and chromosomal abnormalities, spontaneous abortions or congenital malformations. A statistically significant association was observed between the time worked, which was greater in those exposed, and chromosomal alterations. Level of evidence SIGN: 2- because it is a cohort study with a high risk of confounding, bias or chance and a significant probability that the relationship is not causal, since it does not take into account other occupational or lifestyle-related risk factors capable of causing these reproductive alterations. Grade of recommendation SIGN: No recommendation.

Study 6 [37]. Title: “Effect of exposure to inhalational anesthetics on reproductive outcomes and its predictors among healthcare workers in Jimma zone public hospitals: A Comparative Cross-Sectional Study”. Authors: Olika MK, Dessalegn ZM, Mekonin GT, Aboye MB, Wedajo MB, Ilala TT, Abebe DM and Demissie WR. Journal and year of publication: *International Journal of Women’s Health*, 2022. Country and language: Ethiopia, English. Objective: To assess the reproductive alterations produced by inhalational anesthetics in health professionals. Type of study: Retrospective cohort design. Study population: 483 Ethiopian health professionals. Data collection method: Self-completed survey on work history and the occurrence of spontaneous abortions or congenital malformations. The exposed cohort consisted of 146 health professionals working in the operating room, and the unexposed cohort consisted of 146 professionals working in hospital outpatient clinics. Type of inhalational anesthetic: Not specified. Results: A total of 292 professionals (60.4%) responded to the survey, including 57 exposed and 57 non-exposed women. A higher risk of spontaneous abortions and congenital malformations was observed in exposed women. A statistically significant association was found between the time worked, which was greater in exposed women, and the occurrence of spontaneous abortions or congenital malformations. Level of evidence SIGN: 2- because it is a cohort study with a high risk of confounding, bias or chance and a significant probability that the relationship is not causal, since the response rate is low, and it does not take into account other occupational or lifestyle-related risk factors capable of causing these reproductive alterations. Grade of recommendation SIGN: No recommendation.

## 4. Discussion

Although all the studies selected for this review have a retrospective cohort design, they present important methodological differences, especially in the way of calculating the degree of exposure. Most of the studies analyzed [16,33,34,36,37] calculate the degree of exposure based on the time worked in an environment where inhalational anesthetics could be used and do not evaluate, as would be more appropriate, their environmental concentration and the individual exposure of each worker [37]. This is an essential issue in order to properly classify the study subjects within the cohort of exposed or unexposed healthcare professionals and, thus, to be able to determine the existence of a causal relationship between occupational exposure to inhalational anesthetics and the teratogenic effects that appear in pregnant healthcare professionals who are in contact with them, effects that could manifest themselves in an increase in spontaneous abortions or congenital malformations.

The studies analyzed that find an association between the use of inhalational anesthetics and spontaneous abortions or congenital malformations [34,36,37] have a lower level of SIGN evidence and lack a SIGN grade of recommendation. This is because they present a high possibility of confounding bias by not taking into account other occupational or lifestyle risk factors to which healthcare professionals are also subjected and may result in spontaneous abortions or congenital malformations. These factors include exposure to antineoplastic drugs, ionizing radiation or sterilizing agents; the consumption of coffee, alcohol or tobacco; age; stress; jobs involving heavy lifting; long working hours; and highly variable schedules that prevent adequate rest [34,36,37]. They also present confounding biases because the cohorts of exposed and unexposed healthcare professionals perform their work in very different working conditions, such as operating rooms and hospital outpatient clinics, which do not differ exclusively in the use of inhalational anesthetics, making it difficult to establish a causal relationship between their use and the occurrence of spontaneous abortion or congenital malformations [36,37]. In addition, two of these studies have a high nonresponse bias [34,37], since a large percentage of the healthcare professionals surveyed did not respond to the questionnaire, which may lead to erroneous results, since the sample analyzed may not be representative of the study population.

The study by Borayek et al. [36] quantifies urinary isoflurane levels without finding an association with the occurrence of spontaneous abortions, congenital malformations or chromosomal alterations, which could indicate that the increase in these adverse effects of pregnancy could be due to causes other than inhalational anesthetics.

It is worth noting that the studies by Borayek et al. [36] and Olika et al. [37] show that working time was greater in the cohort of exposed health professionals, and even the study by Nagella et al. [34] found a statistically significant association between working time and the appearance of congenital malformations, which could indicate that this could be an important causal factor in the origin of these anomalies.

It should be noted that the studies by Nagella et al. [34] and Borayek et al. [36] indicate that most operating rooms analyzed did not use gas extraction systems (scavenging systems). This shortcoming, which, in principle, may suggest that a high exposure to inhalational anesthetics may be the cause of these problems, may also be indicative of poorly equipped operating rooms that would allow for overexposure to other teratogenic factors (radiation, sterilizing substances, etc.) that could be the real causes of these reproductive disorders due to the lack of adequate protection [28,29,38,39].

The studies that do not find an association between the use of inhalational anesthetics and the occurrence of spontaneous abortions [16,33,35] or congenital malformations [33] have a higher level of evidence SIGN and present a grade of recommendation SIGN. This is because they present a lower possibility of confounding bias by having selected cohorts of exposed and unexposed healthcare professionals with the same [16] or similar working conditions, such as operating rooms and intensive care units [33,35], which differ mainly in the degree of exposure to inhalational anesthetics. These studies can also avoid confounding bias by taking into account other risk factors to which healthcare professionals are also subjected, such as occupational or lifestyle risk factors, which are themselves capable of producing spontaneous abortions or congenital malformations. These factors include exposure to antineoplastic drugs, radiation or sterilizing agents; the consumption of coffee, alcohol or tobacco; age [16]; stress; jobs that require heavy lifting; long working hours; and highly variable schedules that prevent adequate rest [33].

Although the studies by Lawson et al. [16] and Allweiler et al. [33] do not accurately determine the level of individual exposure of each healthcare professional to inhalational anesthetics [16,33], the study by Eftimova et al. [35] with a higher level of evidence SIGN does quantify the exact exposure of each worker to nitrous oxide, so their results should be carefully considered.

Although with the available evidence it cannot be confirmed that inhalational anesthetics do not cause teratogenic effects, it can be observed that the studies with the highest level of evidence SIGN [16,33,35] found no association between the occupational exposure of health professionals to anesthetic gases and the risk of spontaneous abortion or congenital malformations, results that coincide with the majority of previous reviews [21,23,24,27]. However, in the event that inhalational anesthetics could have an adverse reproductive effect, and according to the results of the studies analyzed, it would be sufficient for operating rooms to have adequate ventilation and a gas extraction system (a scavenging system) to nullify this risk.

In view of the above, it is proposed that, if future research on this subject is conducted, the individual exposure of healthcare professionals to inhalational anesthetics should be accurately quantified, the study should be carried out in operating rooms with adequate gas extraction systems (scavenging systems) and any other occupational or lifestyle factors that may be directly related to reproductive alterations and to which professionals using these drugs are also exposed should be taken into account.

## 5. Conclusions

For the most part, studies evaluating the teratogenic effects of inhalational anesthetics are scarce and have a low level of scientific evidence. The studies with the highest level of evidence do not find an association between the use of inhalational anesthetics and spontaneous abortion or congenital malformations. Therefore, based on these results, it could be deduced that the use of anesthetic gases, especially with gas extraction systems (scavenging systems) and the adequate ventilation of operating rooms, is not associated with the occurrence of spontaneous abortion or congenital malformations.

## Figures and Tables

**Figure 1 healthcare-11-00883-f001:**
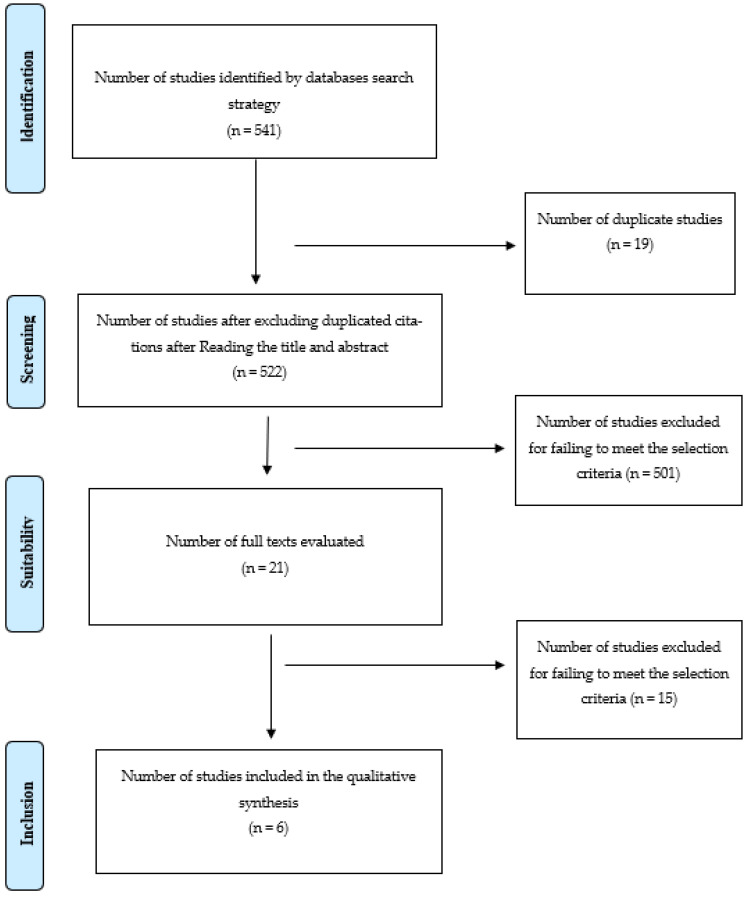
Flow diagram obtained based on the PRISMA statement. Source: adapted by the authors.

**Table 1 healthcare-11-00883-t001:** Comparison of selected studies.

Study	Type of Study	Study Population	Data Collection Method	Type of Inhalational Anesthetic	Results	SIGN Evidence/Recommendation
Lawson et al. [16]	Retrospective cohorts	7842 nurses exposed: working with inhalational anesthetics more than 1 h per day	Questionnaire on work history and occurrence of spontaneous abortions	Halothane, isoflurane, enflurane and nitrous oxide	No increased risk of spontaneous abortion in exposed nurses	2+/C
Allweiler et al. [33]	Retrospective cohorts	Female veterinarians 295 exposed: working in anesthesia 86 non-exposed: working in intensive care	Questionnaire on work history and occurrence of spontaneous abortion or congenital malformations	Not specified	No increased risk of spontaneous abortion or congenital malformations in exposed veterinary females	2+/C
Nagella et al. [34]	Retrospective cohorts	345 female anesthesiologists exposed: working in the operating room during the first trimester of gestation Not exposed: working outside the operating room during the first trimester of gestation.	Questionnaire on work history and occurrence of spontaneous abortion or congenital malformations	Isoflurane, halothane, desflurane, sevoflurane and nitrous oxide	Increased risk of spontaneous abortions in exposed women Association between time worked and congenital malformations No association between type of inhalational anesthetics and spontaneous abortions or congenital malformations	2-/ No recommendation
Eftimova et al. [35]	Retrospective cohorts	23 exposed workers: working in operating room 20 non-exposed workers: working in intensive care	Questionnaire on work history and the occurrence of spontaneous abortion Measurement of nitrous oxide concentration in the operating room	Nitrous oxide	No increased risk of spontaneous abortions in exposed women No association between nitrous oxide concentrations and spontaneous abortions	2++/B
Borayek et al. [36]	Retrospective cohorts	32 exposed nurses: working in the operating room more than 6 h per day for 6 days per week 32 non-exposed nurses: working in hospital outpatient clinics	Questionnaire on work history and the occurrence of miscarriages or congenital malformations Measurement of isoflurane concentration in urine Evaluation of chromosomal alterations	Isoflurane	Longer working time in exposed women Higher risk of spontaneous abortions, congenital malformations and chromosomal alterations in exposed women No association between isoflurane in urine and spontaneous abortions, congenital malformations or chromosomal alterations Association between time worked and chromosomal alterations	2-/ No recommendation
Olika et al. [37]	Retrospective cohorts	146 exposed health professionals: working in the operating room 146 non-exposed health professionals: hospital outpatient clinics	Questionnaire on work history and occurrence of spontaneous abortion or congenital malformations	Not specified	Longer working time in exposed women Increased risk of spontaneous abortions and congenital malformations in exposed women Association between time worked and spontaneous abortion or congenital malformations	2-/ No recommendation

## Data Availability

The data used to support the findings of this study are available from the corresponding author upon request.

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
