# Peer review of "Occupational Exposure to Inhalational Anesthetics and Teratogenic Effects: A Systematic Review"

_healthcare, 2023, doi:10.3390/healthcare11060883_

Round 1

Reviewer 1 Report

The authors conducted an interesting systematic review and it may be viable for publication in HealthCare as long as it addresses the following comments:

- The authors comment that the evidence found is of low quality to demonstrate the teratogenic effect of gases in health personnel. They mention that the studies found are of low quality to demonstrate a causal relationship. However, in your conclusion to ensure that this association does not exist, which is confusing, since the studies found are of low quality, how can we determine the lack of association? It is suggested to change the wording according to the evidence found.

- High-quality studies could be recommended to answer, with a high level of scientific evidence, the question posed.

- We consider that, due to the level of evidence found, the question would remain unanswered.

Taking into account clinical considerations, your evaluation of the methodology used, and the statistical power of the study, do you think there is clear evidence of an association between exposure and outcome?

Are the results of this study directly applicable to the patient group targeted in this guideline?

- Several errors are observed in English.

Author Response

Thank you very much for the recommendations and suggestions. They have guided us to improve the quality of the manuscript and we have tried to follow them with the maximum rigor.

Reviewer 2 Report

This is an important and comprehensive study A considerable amount of poor scientific information has been published on this somewhat emotive subject. The authors review has contributed to a full and comprehensive understanding of the subject.

There are two points which need attention.

1.There is reference throughout the paper to "anaesthetic gases whereas ,infact, there is only ONE anaesthetic gas which is Nitrous Oxide.The other agents are volatile liquids which vaporise and should be referred to as vapours throught the text.

2.The correct term for the removal of gases and vapours from anaesthetic circuits ,in order, to reduce atmosheric pollution/contamination is "cavenging" and this term should be used throughout the manuscript.

Author Response

(The authors gave the same response as above.)
